# Biofortification of Hens Eggs with Polyunsaturated Fatty Acids by New Dietary Formulation: Supercritical Microalgal Extract

**DOI:** 10.3390/ani10030499

**Published:** 2020-03-17

**Authors:** Izabela Michalak, Marita Andrys, Mariusz Korczyński, Sebastian Opaliński, Bogusława Łęska, Damian Konkol, Radosław Wilk, Edward Rój, Katarzyna Chojnacka

**Affiliations:** 1Department of Advanced Material Technologies, Wrocław University of Science and Technology, Smoluchowskiego 25, 50-372 Wrocław, Poland; izabela.michalak@pwr.edu.pl (I.M.); radekwilk@gmail.com (R.W.); katarzyna.chojnacka@pwr.edu.pl (K.C.); 2Department of Environment Hygiene and Animal Welfare, Wrocław University of Environmental and Life Sciences, Chełmońskiego 38C, 51-630 Wrocław, Poland; marita.swiniarska@upwr.edu.pl (M.A.); mariusz.korczynski@upwr.edu.pl (M.K.); sebastian.opalinski@upwr.edu.pl (S.O.); 3Faculty of Chemistry, Adam Mickiewicz University in Poznan, Umultowska 89b, 61-614 Poznań, Poland; bogunial@amu.edu.pl; 4Supercritical Extraction Department, Łukasiewicz Research Network—New Chemical Syntheses Institute, Aleja Tysiąclecia Państwa Polskiego 13a, 24-110 Puławy, Poland; edward.roj@ins.pulawy.pl

**Keywords:** Functional food, biofortified eggs, dietary feed supplement, supercritical fluid extraction, *Spirulina platensis*, algae extract

## Abstract

**Simple Summary:**

Polyunsaturated fatty acids (PUFAs) are essential components of the human diet. The consumption of PUFA can be increased by enriching animal products, such as eggs, with bioactive compounds. New dietary feed supplements were elaborated to increase the level of PUFAs in hens eggs. Nutritional and technological aspects related to the development of functional eggs were discussed. Experimental works dealing with feeding hens new supplements and an investigation of the bioavailability of PUFAs and their transfer to eggs content were undertaken. Analysis of the results obtained during tests showed that used additives may increase the content of such fatty acids (FAs) in eggs: docosapentaenoic acid (DPA, C22:5 n-3), eicosadienoic acid (C22:2 n-6), and the total omega-6 polyunsaturated fatty acids content (PUFA n-6). The investigated additives may also reduce the content of saturated fatty acids (SFA) such as pentadecanoic acid (C15:0).

**Abstract:**

The aim of the study was to evaluate the effect of *Spirulina platensis*, formulation containing microalgal extract, post-extraction residue, and formulation without algal extract (containing only emulsifier) on the content of FAs in the eggs of laying hens. The experiment was conducted on 90 laying hens (ISA Brown) as a completely randomized design. Hens were assigned to five experimental groups (six replicates). The FAs content in eggs was determined after 30, 60, 90, and 120 days of the experiment. There were no statistically significant differences in FA profiles after 30 days of the experiment. It was shown that after 60, 90, and 120 days of the experiment, the investigated additives had a significant impact on the content of such acids as: dodecanoic acid (C12:0), C15:0, nonadecanoic acid (C19:0), myristoleic acid (C14:1 n-5), α-linolenic acid (ALA, C18:3 n-3), DPA, C20:2 n-6, and decosahexaenoic acid (DHA C22:6 n-6). There were also significant differences in total PUFA n-3, PUFA n-6, and n-6/n-3 ratio in eggs. The obtained results suggest that the use of algae extract and emulsifier in laying hens nutrition has the greatest impact on the FA profile in the eggs.

## 1. Introduction

Recently, a market demand for a new generation of functional food with pro-health benefits has been identified. Consumers have become conscious of the relation of diet to their health. Not only the knowledge about the caloric content of food, but the fact that it provides all the necessary ingredients is now the most important. This created a new niche for products with increased levels of biologically active compounds [1]. An example of such compounds sought after by consumers are PUFAs, mainly due to their health benefits (development and maintenance of brain and vision, protection against cardiovascular diseases) [2].

The market offers many products that are industrially fortified with PUFA n-3 and PUFA n-6 (which differ in the location of the first double bond between the carbon atoms), e.g., oils, mayonnaises, margarines, salad dressings, bakery products, infant formula, milk, meat products, farmed fish, and eggs [3]. Many products fortified with PUFAs origin from poultry production and especially enriched eggs are becoming attractive for the human diet. The poultry diet (chicken/hens) is mainly supplemented with additives rich in PUFAs, for example sardine oil [4], menhaden oil [5], rapeseed/corn oils [6], linseed oil and fish oil [7], flaxseed [8]. However, the main disadvantage of the eggs enriched with FA from, for example, fish oil, is the risk of their oxidation and decrease in PUFA n-3 contents during storage [5].

Other natural source rich in PUFAs, as well as antioxidants are for example marine algae: *Sargassum dentifebium* [9], *Macrocystis pyrifera*, *Enteromorpha* spp., and *Sargassum sinicola* [4], marine algae (different species) [5] or microalgae [7]: *Chlorella* spp. [10], *Chlorella fusca, Phaeodactylum tricornutum*, *Isochrysis galbana* [11], *Schizochytrium* sp. [8,12], *Nannochloropsis oculata* [6], and *Nannochloropsis gaditana* [13]. It is worth noting that the use of biomass of microalgae (*Spirulina maxima*) in laying hens diet has already been evaluated by some of the authors of this paper [14].

Usually, poultry diet is based on the standard feeding mixture which is composed of raw materials being rich in PUFAs [15]. In this paper, we propose to use algae in the form of extract added to drinking water. This approach has not been studied so far. Extracts are a concentrated form of biologically active compounds that should be easily bioavailable to animals [16]. For the extraction of active compounds (including PUFAs), we used microalgae *Spirulina* (*Arthrospira*) *platensis*. This alga is produced commercially in large outdoor ponds under controlled conditions or obtained directly from lakes. It is estimated that the current worldwide production of *Spirulina* is about 3000 metric tons per year. In the last 30 years, it was mainly used for food and specialty feeds [17].

*Spirulina* sp. serves as a rich source of proteins (higher content than in chicken egg or meat, beef meat, skimmed powdered milk, beer yeast, cheese, fish), vitamins, minerals and pigments [17]. The concentration of lipids in this microalga was examined and it was stated that *Spirulina* contains 6%–13% lipids, half of which can be expressed as total FAs [18]. Zinnai et al. [19] reported that *Spirulina platensis* is characterised by a lipid fraction with a high proportion of PUFAs such as gamma-linoleic acid (GLA C18:3, n-6). The composition of PUFA in *Spirulina* sp. depends on many factors, including the environment, growth phase [20], temperature, and composition of a culture medium [21].

Supercritical fluid extraction (SFE) of biologically active compounds from *Spirulina* sp. was studied. Mendes et al. [22] performed SFE of GLA from *Spirulina maxima* with CO_2_ and in the second variant, with addition of a polar co-solvent (ethanol)_._ The addition of ethanol increased the extraction yield of lipids, among others GLA, which was in the glycolipid fraction. Moreover, the extraction yield of GLA also increased with the temperature and pressure. These results were confirmed by Quihui [23], who performed SFE of *Spirulina platensis*. Andrich et al. [24] extracted oils (mainly GLA) from *Spirulina platensis* in different experimental conditions (pressure 250, 400, 550, and 700 bar and temperatures 40 and 55 °C, 4 h). The highest tested pressure and temperature were the most efficient in the extraction of lipids. It is suggested, in the available literature, that algal extracts can be utilized as a direct source of dietary FAs and may be an alternative to currently available sources of FAs, for manufacturers of products rich in PUFA n-3 and PUFA n-6 [5].

Algae biomass is a rich source of FAs. However, it contains fiber that can be a limiting factor in poultry nutrition. Therefore, it is better to use algae extracts. Due to the fact that the extracts have a liquid form, it seems that it is better to use them as supplements for a drinking water which was possible thanks to the investigated formulation. In addition, the extract used as an additive to feed could be exposed to high temperatures during its production, which could have a negative impact on the biologically active compounds contained in it. Emulsifiers are added to increase the solubility of the extracts in water. However, emulsifiers can affect the results of the experiment. The SFE process also produces a large amount of residue, which can also contain biologically active compounds.

Therefore, the aim of the present study was to evaluate the effect of the formulation of SC-extract, microalgae *Spirulina platensis*, post-extraction residue, and formulation without extract (containing only an emulsifier, as a positive control) on the performance of laying hens, content of FAs in the raw materials, and in the egg yolk.

## 2. Materials and Methods

### 2.1. Chemicals

All used chemicals were of analytical grade.

#### 2.1.1. Feedstock—*Spirulina* Lyophilizate

WB Im-und Export W. Beringer & Co. GmbH (Görmin, Germany) provided *Spirulina platensis* which was used in the present study. The composition of the biomass was given by the supplier. It was dark green fine powder with particle size < 80 mesh. The characteristics of the biomass was as follows: moisture < 8.0%, ash < 10.0%, protein > 60.0%, toxic metals (Pb < 3.0, Cd < 1.0, Hg < 0.1, As < 3.0 mg kg^−1^). In terms of microbiological composition, the biomass was characterized by: total viable count < 150,000 CFU (Colony Forming Unit) g^−1^, yeasts and moulds < 100 CFU g^−1^, *Enterobacteriaeae* < 100 CFU g^−1^, *Escherichia coli*, *Staphylococus aureus* negative in 1 g, *Salmonella* negative in 10 g. *Spirulina* was free from pesticides and preservatives.

#### 2.1.2. Supercritical Fluid Extraction of the Biomass with CO_2_ (SFE-CO_2_)

Microalga *Spirulina platensis* was subjected to the extraction with carbon dioxide (without addition of co-solvent) at Supercritical Extraction Department in Łukasiewicz Research Network—New Chemical Syntheses Institute in Puławy. The loading capacity of installation was about 1 L. It can operate at pressures up to 1000 bar and extraction temperature up to 200 °C, with a maximum CO_2_ flow rate up to 15 L·h^−1^. Detailed information has been presented by Rój [25]. The experimental conditions were as follows: 700 bar, temperature 40 °C, extraction time 8 h. The extraction yield was 5.6% from 1 kg of raw biomass of *Spirulina platensis*, so 56 g of the extract was obtained and 903 g of post-extraction residues. After the extraction process, the extract was stored in tightly closed dark glass bottles to avoid oxidation [26].

#### 2.1.3. Algal Extract Formulation

Algal extract from *Spirulina platensis* obtained by SFE-CO_2_ (SC-extract) was used as the component of the formulation designed for feeding experiments on laying hens. The composition of the preparation is presented in Table 1. The final formulation was easily soluble in drinking water for hens.

#### 2.1.4. Animal Diets and Experimental Design

The feeding experiments on laying hens were approved by the Second Local Ethical Committee on Animal Testing at Wrocław University of Environmental and Life Sciences (No 5/2015, 21 January 2015). The trials were performed at the research station of Wrocław University of Environmental and Life Sciences in Swojec (Poland).

A total of 90 laying hens (ISA Brown, 36 weeks of age) were housed in a 3-tier battery system, in a room with a controlled climate and light regimen of 16L:8D. The initial laying index, before the beginning of experiment, was on the level of 91%. The feeding experiment was conducted for 120 days. The basal diet (Table 2) was formulated according to the nutrient recommendations for laying hens [27]. The laying hens were fed *ad libitum* and water was provided by nipple drinkers (volume of 1 L) separately for each cage.

The completely randomized design was applied to the feeding experiment. Birds were divided into 5 groups (6 replicates per group, 3 hens in each replicate). The cages were located on the right and left side of the battery system. Each replication consisted of cages from the first, second and third floor on the right and left side of the battery system. This layout best reflects conditions in the room. The division of groups was as follows: (1) Control—fed with basal diet and drinking water, (2) Formulation of SC-extract—fed with basal diet and 0.2% microalgal extract in drinking water; (3) Microalgae—fed with basal diet containing 15 g kg^−1^ of microalgae *Spirulina platensis* and drinking water; (4) Residue—fed with basal diet containing 15 g kg^−1^ of post-extraction residues and drinking water; (5) Formulation without extract (containing only an emulsifier—positive control) fed with basal diet, and 0.2% formulation without extract in drinking water. When choosing the doses of extracts that were investigated during the study, the economic criterion was mainly taken into account, i.e., whether the cost of producing the additive (costs of obtaining the raw material, its extraction, and formulation production) would be rational for farmers.

#### 2.1.5. Performance of Laying Hens

Throughout the experiment, all groups were monitored for egg production. Eggs were collected and weighed daily. Eggs production was determined by dividing the number of eggs laid over the course of the experiment by the number of hens in the same period (expressed as percentage of egg production). Feed intake was recorded once per week. Feed conversion ratio (FCR) was calculated by dividing the feed intake by the mass of eggs.

### 2.2. Sample Collection and Storage of Eggs

Six randomly collected eggs from each replicate after 30, 60, 90, and 120 days of the experiment in all the examined groups were analyzed. Eggs were broken in order to separate albumen, yolk and shell. Egg yolks from each replicate (*n* = 6) were mixed homogeneously in order to obtain representative samples. Then, samples were kept in the temperature of −40 °C. In the next step, the samples underwent lyophilization using Free Zone 18 Liter Console Freeze Dry System (Labconco Corporation, Kansas City, MO, USA).

### 2.3. Preparation of Samples for FAs Analysis

#### 2.3.1. Lyophilized Eggs

Extraction of lipids from lyophilized eggs (50 mg) was performed according to the Folch method using 10 mL of the mixture chloroform:methanol (2:1; *v v*^−1^). Extraction was performed by 30 min providing intensive mixing. Then the solvent from the collected solution (2 mL) was evaporated in a rotary evaporator at a temperature < 50 °C until constant weight. In the next step, 0.5 mL of tert-butyl methyl ether (MTBE) was added to the dry residue. Subsequently, to the mixture of 0.25 mL of 0.25 M solution of trimethylsulfonium hydroxide in methanol as a derivatizing agent was added. Then, the mixture was stirred for 5 min in a room temperature. Finally, the internal standard (solution of 10 µL of methyl undecanoate, 85 mg in 5 mL MTBE) was added.

#### 2.3.2. Formulation of SC-Extract

The samples (0.5 mL, *n* = 2) of the extract were evaporated in a rotary evaporator at a temperature < 50 °C until constant weight. We dried the solid residue under vacuum per 10 min which was then dissolved in 0.5 mL of MTBE. In the next step, 0.25 mL of 0.25 M solution of trimethylsulfonium hydroxide in methanol was added and after 5 min of mixing at room temperature, the internal standard (20 µL of methyl undecanoate—85 mg in 5 mL MTBE).

#### 2.3.3. Formulation Without SC-Extract

Bredol 694 (Emulsifier) (approximately 25 mg, *n* = 2) was dissolved in 0.5 mL of MTBE. Then, the procedure was the same as for the formulation of SC-extract.

#### 2.3.4. Microalga—*Spirulina Platensis* and Post-Extraction Residue

The samples (250 mg, *n* = 2) were extracted with 2 mL of MTBE in the temperature of 70 °C for 2 h. Then, after cooling the sample, the internal standard (20 µL of methyl undecanoate—85 mg in 5 mL MTBE) was added and the obtained solution was filtered. To 0.5 mL of the solution derivatizing agent, 0.25 mL of 0.25 M solution of trimethylsulfonium hydroxide in methanol was added.

### 2.4. Fatty Acids Analysis in Samples

A Varian 450-GC gas chromatograph was used to determine fatty acids as methyl esters. The chromatograph operated on the following parameters: split: 1:50; injector temperature: 250 °C; carrier gas: helium, flow rate 1 mL min^−1^; Agilent HP-INNOWax GC column, 30 m × 0.53 mm, 1.0 µm film thickness; temperature program: 50 °C isothermal for 2 min; linear gradient of 10 °C min^−1^ to 250 °C (20 min), isothermal 240 °C for 22 min; detector: a flame ionization detector (FID), 250 °C. To the injection port, 1 µL of the prepared sample was introduced. The retention times of standards was used to identify methyl esters of FAs.

### 2.5. Statistical Methods

The software *Statistica* ver. 10 was used for the statistical elaboration of the obtained results. Shapiro-Wilk test was applied to assess the normality of the distribution of experimental results and to select an appropriate statistical test in order to investigate the statistically significant differences between the tested groups. In the case of normal distribution, a one-way analysis of variance (ANOVA) with Tukey test was used to investigate the differences between groups. The Kruskal–Wallis test was applied, if the distribution was not normal. Effects were considered significant at a probability of *p* < 0.05 and *p* < 0.01.

## 3. Results and Discussion

### 3.1. Content of FAs in The Raw Materials

The FAs can be mainly extracted from algal biomass using Soxhlet apparatus with organic solvent or SFE-CO_2_ as a solvent [29,30,31]. It was noticed that SFE favored isolation of saturated and unsaturated fatty acids when compared to traditional extraction techniques with the use of organic solvents. Many studies showed that SFE is characterized by a higher efficiency of extraction of biologically active substances from a natural source when compared to other extraction techniques [31,32]. Table 3 presents the percentage weight of FA in dry matter of the samples that were used in laying hens feeding, i.e., microalga *Spirulina platensis*, formulation of SC-extract from *S*. *platensis*, post-extraction residue of *S*. *platensis,* and the formulation that was used to prepare the application form of supercritical extract.

The formulation of SC-extract was the main source of FAs, especially 16:0, 16:1 (n-7), 18:0, 18:1 (n-9), 18:2 (n-6) and 18:3 (n-6) when compared with other examined samples (microalgae, residue, formulation without SC-extract). 16:0 was the major FA. These observations are in agreement with the data reported by Andrich et al. [24]. In the SC-extract from *S. platensis*, the dominating FAs were 16:0 (40% of total FA by weight at 700 bar, 55 °C) > 18:3 (n-6; 20.2%) > 18:2 (n-6; 15.0%) > 18:1 (n-9; 9.9%). These values were much higher than in the present study, however, we examined the formulation of SC-extract in which the SC-extract constituted 1.34%, not the raw extract.

In the present work, the main FAs, in the biomass of *S*. *platensis* before extraction process were 16:0> 18:2 (n-6) > 18:3 (n-6) >18:1 (n-9)> 16:1 (n-7) = 18:0. Similar FA present in the *Spirulina maxima* were reported by Batista et al. [33]: 16:0 (1078 mg 100g^−1^) > 18:2 (n-6; 481 mg 100 g^−1^) > 18:3 (n-6; 452 mg 100 g^−1^) > 16:1 (189 mg 100 g^−1^) > 18:1 (115 mg 100 g^−1^) > 18:0 (32 mg 100 g^−1^). Post-extraction residue and the formulation without SC-extract contained low levels of FAs. On the basis of the preliminary analysis it was supposed that the formulation of SC-extract will be the main source of FAs accumulated in the eggs of laying hens. It is worth noting that relatively large amounts of the 16:0 and palmitoleic acid 16:1 were found in the SC-extract. These FAs, with strong anti-free radical and antioxidative properties, rarely occur in algae [34]. In SC-extracts, they become a valuable additive to animal feed.

### 3.2. Production Parameters of Laying Hens

Results on the production parameters of the tested animals are shown in Table 4.

The obtained results did not show any significant positive influence of the investigated additives on the laying rate. However, birds receiving the formulation of SC-extract were characterized by the highest egg production after the 30th, 60th and 90th days of the experiment. Based on the obtained results, it can be concluded that the FCR of hens receiving microalgae was significantly (*p* < 0.05) lower when compared to hens receiving formulation without SC-extract. The FCR of hens receiving microalgae was also the lowest compared to other experimental groups, however, this was not a statistically significant result. Based on the results, it can be concluded that the used additives had no significant effect on laying hen performance. Other results were obtained by Mariey et al. [35], who found that the enrichment of the hens’ diet with *Spirulina platensis* significantly improved egg production and FCR. Zahroojian et al. [36] failed to demonstrate that the use of *Spirulina platensis* significantly affected egg production, feed intake, and FCR. Other studies of Zahroojian et al. [37] also did not show a significant effect of the use of *Spirulina platensis* in the diet of laying hens on the performance of birds.

### 3.3. Content of Fatty Acids in Eggs Yolks

The results regarding the content of FAs after 30, 60, 90, and 120 days of the experiment are presented in Table 5, Table 6, Table 7, and Table 8 respectively.

Analysis of the results did not show a significant effect of the used additives on the FA profile after 30 days of the experiment.

After 60 days of the experiment, the eggs of hens from the groups receiving the formulation of SC-extract and formulation without SC-extract were characterized by a significantly higher content of DPA compared to hens from the control group and group receiving the addition of microalgae.

After 90 days of the experiment, the eggs of hens from the group receiving residue were characterized by a significantly higher (*p* < 0.01) content of 14:1 (n-5) compared to the group receiving formulation without SC-extract. Eggs of hens from the group receiving the residue and the addition of microalgae were characterized by a significantly (*p* < 0.01) lower content of ALA compared to other groups. Eggs from the control group were characterized by a significantly (*p* < 0.01) lower content of 19:0 compared to group receiving the residue. In the eggs of hens from the control group and those receiving the formulation of SC-extract, a significantly lower (*p* < 0.01) C20:2 (n-6) content was noted compared to the group receiving the addition of microalgae. Eggs of hens from the group receiving the addition of microalgae were also characterized by a significantly higher (*p* < 0.05) content of 20:3 (n-3) compared to the group receiving formulation without SC-extract. Eggs of hens from the control group and these receiving formulation of SC-extract and formulation without SC-extract were characterized by a significantly higher (*p* < 0.01) content of DHA compared to the groups receiving the addition of microalgae and residue. Eggs of hens from the control group and those receiving formulation of SC-extract were also characterized by a significantly higher (*p* < 0.01) content of PUFA n-3 compared to the groups receiving the addition of microalgae and residue. Microalgae and residue groups were also characterized by a significantly lower (*p* < 0.05) content of PUFA n-3 compared to group receiving formulation without SC-extract. In eggs from the group receiving the microalgae, a significantly higher (*p* < 0.05) content of PUFA n-6 was found compared to the control group and group receiving formulation of SC-extract. The ratio of n-6/n-3 acids was significantly lower (*p* < 0.01) in the eggs of hens from the control group, formulation of SC-extract group, and formulation without SC-extract group compared to the groups receiving microalgae and residue.

After 120 days of the experiment, it was shown that eggs from group receiving the formulation without SC-extract had significantly less (*p* < 0.01) 12:0 than in eggs from the control group. Eggs from the group receiving formulation without SC-extract were also characterized by a significantly lower (*p* < 0.05) content of 12:0 compared to the microalgae and residue groups. Eggs from the control group were characterized by a significantly higher (*p* < 0.05) content of 15:0 compared to the group receiving the addition of microalgae. Eggs from the control and residue groups were characterized by a significantly lower (*p* < 0.05) content of DPA than microalgae group. Control eggs showed a significantly higher PUFA/SFA ratio (*p* < 0.01) than eggs from the residue group.

It is difficult to compare the obtained results with the literature data, because the effect of algal extracts added to the drinking water has not been studied so far. More data concern the effect of algae on the content of FAs in eggs, for example [4,8,9,12,13]. Šefer et al. [12] found that the inclusion of algal product DHA Gold^®^ from *Schizochytrium* spp. (at the optimal dose 10 g kg^−1^ of feed among 5 and 7 g kg^−1^ of feed) led to significant increases of DHA, but not EPA, content in eggs. Bruneel et al. [13] also found that EPA derived from microalga *Nannochloropsis gaditana* was hardly accumulated in yolk lipids. It preferentially converted to DHA and deposited in yolk phospholipids. For the algal dose of 10%, the level of DHA in egg yolk was doubled when compared to the eggs from the control group fed with a standard feed.

The results obtained in the present study show that the algal extract is a better source of FAs than the raw biomass of microalgae. However, this solution is much more expensive due to the costs of SFE. Because of this fact, Colla et al. [21] suggested direct consumption of *Spirulina* as a nutritional supplement instead of extract. On the other hand, the addition of the microalgal SC-extract to the diet is very relatively low (in this study 1.3%) and assures better enrichment of eggs with FAs. Some authors indicate that higher content of algae in the diet did not increase the content of some FA. For example, Al-Harthi and El-Deek [9] examined the effect of different contents (0%, 3%, and 6%) of brown marine algae (*Sargassum dentifebium*) on yolk lipid profiles in egg yolks. It was shown that higher contents of algal biomass in the diet caused a decrease in the content of FAs in eggs. A significant increase in 18:1 (n-9) content was observed when feed containing 3% of algae was supplied to the hens’ diet, while the content of this FA decreased with an increasing dose of algae. Moreover, the inclusion of either 3% or 6% of algae resulted in a significant decrease of the content of 18:2 (n-6).

During the feeding experiment, the n-3/n-6 ratio was also changed. The appropriate ratio of n-6/n-3 acids should be 4–5/1 [38]. After day 30 of the experiment, the best ratio of n-3/n-6 (5.98/1) was characterized by the group fed with the formulation containing residue. The control group had the worst acid ratio (6.52/1). However, these differences were not statistically significant. After 60 days of experiment, the best ratio of FA was characterized by the group fed with the formulation without SC-extract (5.75/1). The worst was the group receiving the addition of microalgae (6.17/1), however these differences also were not statistically significant. After 90 days of experiment, the group fed with the formulation of SC-extract was characterized by the best ratio of FAs (7.70/1). The group receiving the microalgal supplement had the worst FA ratio (11.59/1) and it was statistically significant difference (*p* < 0.01). After 120 days of experiment, the best ratio of these acids was characterized by the group fed with the formulation containing residue (8.06/1) while the worst was for the control group (8.61/1)—not statistically significant. In the course of the whole experiment, the best ratio of FA was recorded after 60 days, in a group fed with formulation without SC-extract (5.75/1) and the worst after 90 days of, in microalgae fed group (11.59/1). Generally, the worst FAs ratio was recorded in egg yolks collected after 90 days of the experiment. The results suggest that the consumption of eggs fortified with PUFAs may discourage their daily intake to the recommended standards. A similar ratio of n-6/n-3 (6.5-7.7/1) acids was obtained by Farrel [39] who supplemented hens’ diet with fish oil and a combination of fish and vegetable oils.

## 4. Conclusions

In the present study, new dietary feed supplement for laying hens, with the potential to increase the concentration of PUFAs in eggs was investigated. The obtained results suggest that supplementation of the diet with examined additives has no effect on the performance of laying hens. It can also be stated that the greatest influence on the FA profile in the egg was noted for the SC-extract. However, this profile was similar in the group receiving the addition of emulsifier. Thus, it is difficult to draw definite conclusions. Accordingly, further research should be continued in this direction.

## Figures and Tables

**Table 1 animals-10-00499-t001:** The composition of the applied formulation with *Spirulina* extract (Polish patent application. No P. 412010).

Component	The Content of The Components in The Formulation of SC-Extract (g)
*Spirulina platensis* extract (Active ingredient)	6.70
Bredol 694 (Emulsifier)	25.0
Potassium sorbate (Preservative)	0.250
Citric acid (Antioxidant)	0.500
Water (Solvent)	468

**Table 2 animals-10-00499-t002:** Ingredients and chemical composition of basal diet [28].

**Feed Material**	**Diet (88.4% Dry Matter) %**
Wheat	46.2
Sunflower meal 35% of total protein (TP)	5.0
Triticale 11% TP	20.0
Soybean oil	1.4
Soybean meal 46% TP	14.0
Beans	3.0
Monocalcium phosphate	0.4
Calcium carbonate 38%	9.0
PREMIX—POLF KOMPL DJ-1 KOL 1% *	1.0
**Composition**	**Percentage (%)**
Total protein	16.9
Crude fat	2.7
Crude fiber	3.5
Ash	12.0
Lysine	0.77
Methonine + cystine	0.73
Methionine	0.42
Threonine	0.57
Sodium (Na)	0.17
Calcium (Ca)	3.65
Phosphorus (P)	0.32

***** Composition of premix (content per 1 kg)—vitamins (A (E672 retinol)—1,200,000 IU; D_3_ (E671 cholecalciferol)—250,000 IU; E (3a700 octan all-rac-alfa-tokoferyl)—2500 IU), trace elements (Cu (CuSO_4_·5H_2_O)—1500 mg; Fe (FeSO_4_·H_2_O)—7000 mg; J (KI)—150 mg; Mn (MnO_2_)—10,000 mg; Zn (ZnO)—8000 mg; Se (Na_2_SeO_4_)—20 mg), digestibility enhancers (3-phytaze (E1600)—5000 FTU g^−1^), antioxidants (Ethoxyquin (E324)—8 mg; Butylatedhydroxytoluene (BHT)—55 mg; Butylatedhydroxyanisole (BHA)—5 mg), dyes (Canthaxanthin (E161g)—350 mg, Ethyl ester of beta-apo-8′-carotenic acid (E160f)—250 mg), aromats—mix—200 mg.

**Table 3 animals-10-00499-t003:** Percentage weight of FAs in the samples used in laying hens feeding.

Fatty Acids	% Weight of Fats in Dry Matter of the Samples (*n* = 2)							
	Microalga— *Spirulina Platensis*		Formulation of SC-Extract		Residue		Formulation without SC-Extract	
	Mean	SD	Mean	SD	Mean	SD	Mean	SD
6:00	<0.002	-	0.010	0.000	<0.002	-	n.d.	-
10:0	n.d.	-	<0.002	-	n.d.	-	n.d.	-
12:0	n.d.	-	n.d.	-	n.d.	-	<0.003	-
13:0	n.d.	-	n.d.	-	n.d.	-	<0.003	-
14:0	n.d.	-	0.020	0.000	n.d.	-	<0.001	-
15:0	n.d.	-	0.010	0.000	n.d.	-	n.d.	-
16:0	0.255	0.0353	1.95	0.16	0.060	0.014	0.025	0.007
16:1 (n-7)	0.020	0.000	0.200	0.028	0.010	0.000	n.d.	-
17:0	<0.002	-	0.020	0.000	n.d.	-	n.d.	-
18:0	0.020	0.000	0.305	0.021	0.010	0.000	0.010	0.000
18:1 (n-9)	0.040	0.000	1.12	0.07	0.015	0.007	0.030	0.000
18:2 (n-6)	0.155	0.007	0.925	0.063	0.035	0.021	n.d.	-
18:3 (n-6)	0.065	0.007	0.755	0.078	0.020	0.000	n.d.	-
18:3 (n-3)	n.d.	-	<0.005	-	n.d.	-	n.d.	-
19:0	n.d.	-	n.d.	-	n.d.	-	n.d.	-
20:1 (n-9)	n.d.	-	n.d.	-	n.d.	-	n.d.	-
20:2 (n-6)	n.d.	-	0.010	0.000	n.d.	-	n.d.	-
20:3 (n-3)	n.d.	-	0.015	0.007	n.d.	-	n.d.	-
20:4 (n-6)	n.d.	-	n.d.	-	n.d.	-	n.d.	-
20:5 (n-3)	n.d.	-	0.080	0.014	n.d.	-	n.d.	-
22:5 (n-3)	n.d.	-	n.d.	-	n.d.	-	n.d.	-
22:6 (n-3)	n.d.	-	n.d.	-	n.d.	-	n.d.	-

Abbreviations: n.d.—not detected; 12:0—lauric acid; 14:0—myristic acid; 14:1 (n-5)—myristoleic acid; 15:0—pentadecylic acid; 16:0—palmitic acid; 16:1 (n-7)—palmitoleic acid; 17:0—margaric acid; 18:0—stearic acid; 18:1 (n-9)—oleic acid; 18:2 (n-6)—linoleic acid; 18:3 (n-6)—gamma-linolenic acid (GLA); 18:3 (n-3)—alpha-linolenic acid (ALA); 19:0—nonadecylic acid; 20:1 (n-9)—eicosenoic acid; 20:2 (n-6)—eicosadienoic acid; 20:3 (n-3)—eicosatrienoic acid (ETE); 20:4 (n-6)—arachidonic acid (AA); 20:5 (n-3) eicosapentaenoic acid (EPA); 22:5 (n-3)—docosapentaenoic acid (DPA); 22:6 (n-3) decosahexaenoic acid (DHA).

**Table 4 animals-10-00499-t004:** Production parameters of laying hens during the experiment.

Production Parameters		Control	Formulation of SC-Extract	Microalga—*Spirulina Platensi*s	Residue	Formulation without SC-Extract	SEM	*p*-Value
Laying index (%)	After 30 days	89	96	91	93	90	1.08	0.437
	After 60 days	89	94	93	92	90	1.02	0.494
	After 90 days	87	95	91	90	90	1.00	0.205
FCR (g feed/g egg)	After 30 days	1.6 ^ab^	1.6 ^ab^	1.2 ^a^	1.7 ^ab^	2.2 ^b^	0.09	0.022
	After 60 days	2.2	2.3	2.1	2.3	2.3	0.08	0.843
	After 90 days	2.0 ^ab^	1.9 ^a^	2.0 ^ab^	2.5 ^ab^	2.7 ^b^	0.09	0.027

SEM—standard error of measurements; ^a,b^—statistically significant differences between the groups for individual parameters, at *p* < 0.05.

**Table 5 animals-10-00499-t005:** The content of FAs in eggs (% weight of FAs in the sample) after 30 days of the experiment.

Fatty Aacids	Control	Formulation of SC-Extract	Microalga—*Spirulina Platensis*	Residue	Formulation without SC-Extract	SEM	*p*-Value
12:0	0.0107	0.0108	0.0107	0.0089	0.0106	0.00039	0.983
14:0	0.273	0.264	0.271	0.275	0.259	0.0073	0.853
14:1 (n-5)	0.0447	0.0432	0.0447	0.0464	0.0463	0.00143	0.956
15:0	0.0646	0.0633	0.0613	0.0615	0.0628	0.00181	0.908
16:0	21.3	21.3	21.2	21.5	21.1	0.485	0.947
16:1 (n-7)	2.14	2.45	2.08	2.21	2.17	0.055	0.246
17:0	0.168	0.162	0.164	0.161	0.165	0.0038	0.746
18:0	6.70	6.58	6.85	6.68	6.61	0.138	0.542
18:1 (n-9)	39.4	40.1	39.3	40.1	39.5	0.815	0.747
18:2 (n-6)	13.3	13.2	13.4	12.5	13.1	0.293	0.709
18:3 (n-6)	0.0858	0.0878	0.0873	0.0844	0.0945	0.00222	0.883
18:3 (n-3)	0.673	0.663	0.661	0.645	0.648	0.0156	0.981
19:0	0.0558	0.0576	0.0602	0.0574	0.0601	0.00143	0.852
20:1 (n-9)	0.212	0.223	0.239	0.217	0.233	0.0048	0.411
20:2 (n-6)	0.103	0.102	0.115	0.095	0.105	0.0028	0.198
20:3 (n-3)	0.098	0.104	0.105	0.097	0.096	0.0023	0.616
20:4 (n-6)	1.41	1.42	1.44	1.45	1.47	0.028	0.446
20:5 (n-3)	0.477	0.543	0.514	0.508	0.528	0.0142	0.666
22:5 (n-3)	0.274	0.296	0.313	0.309	0.303	0.0074	0.487
22:6 (n-3)	0.735	0.775	0.754	0.760	0.812	0.0163	0.655
SFA	28.5	28.4	28.6	28.8	28.3	0.630	0.999
MUFA	41.9	42.6	41.7	42.5	41.9	0.868	0.753
PUFA	17.1	17.2	17.4	16.3	17.2	0.367	0.681
PUFA n-3	2.27	2.41	2.37	2.34	2.41	0.044	0.693
PUFA n-6	14.9	14.8	15.1	14.0	14.8	0.324	0.666
n-6/n-3	6.52	6.14	6.36	5.98	6.15	0.065	0.084
PUFA/SFA	0.608	0.607	0.613	0.571	0.611	0.0058	0.118

SEM—standard error of measurements; SFA—saturated fatty acids; MUFA—monounsaturated fatty acids.

**Table 6 animals-10-00499-t006:** The content of FAs in the eggs (% weight of FAs in the sample) after 60 days of the experiment.

Fatty Acids	Control	Formulation of SC-Extract	Microalga— *Spirulina Platensis*	Residue	Formulation without SC-Extract	SEM	*p*-Value
12:0	0.0107	0.0107	0.0107	0.0089	0.0107	0.00040	0.854
14:0	0.285	0.256	0.267	0.280	0.276	0.0075	0.584
14:1 (n-5)	0.0481	0.0398	0.0431	0.0479	0.0481	0.00151	0.270
15:0	0.0550	0.0582	0.0584	0.0564	0.0534	0.00160	0.848
16:0	21.5	21.3	21.2	21.6	21.9	0.485	0.872
16:1 (n-7)	2.25	2.05	2.07	2.34	2.28	0.056	0.255
17:0	0.155	0.164	0.163	0.153	0.152	0.0036	0.608
18:0	6.97	6.85	6.77	6.55	6.88	0.140	0.316
18:1 (n-9)	40.1	40.2	40.3	40.0	39.5	0.810	0.891
18:2 (n-6)	12.3	12.8	13.0	12.4	12.5	0.270	0.633
18:3 (n-6)	0.0799	0.0949	0.0846	0.0930	0.0950	0.00245	0.181
18:3 (n-3)	0.576	0.584	0.663	0.632	0.622	0.0144	0.261
19:0	0.0603	0.0633	0.0574	0.0527	0.0544	0.00148	0.268
20:1 (n-9)	0.216	0.228	0.236	0.217	0.232	0.0046	0.424
20:2 (n-6)	0.091	0.098	0.105	0.218	0.097	0.0149	0.689
20:3 (n-3)	0.098	0.112	0.105	0.099	0.098	0.0023	0.219
20:4 (n-6)	1.47	1.45	1.41	1.35	1.41	0.028	0.493
20:5 (n-3)	0.517	0.548	0.461	0.505	0.538	0.0140	0.317
22:5 (n-3)	0.299 ^a^	0.339 ^b^	0.248 ^Aa^	0.315 ^AaBb^	0.379 ^Bb^	0.0088	0.005
22:6 (n-3)	0.784	0.799	0.767	0.778	0.722	0.0160	0.625
SFA	29.1	28.7	28.5	28.8	29.3	0.630	0.783
MUFA	42.6	42.5	42.6	42.6	42.1	0.861	0.915
PUFA	16.2	16.7	16.9	16.4	16.5	0.343	0.807
PUFA n-3	2.35	2.47	2.36	2.42	2.45	0.048	0.776
PUFA n-6	13.9	14.3	14.6	14.1	14.1	0.301	0.748
n-6/n-3	5.91	5.82	6.17	5.84	5.75	0.051	0.086
PUFA/SFA	0.560	0.586	0.695	0.575	0.566	0.0047	0.118

SEM—standard error of measurements; SFA—saturated fatty acids; MUFA—monounsaturated fatty acids; ^a, b^—statistically significant differences between the groups for individual parameters, at *p*< 0.05.; ^A, B^—statistically significant differences between the groups for individual parameters, at *p* < 0.01.

**Table 7 animals-10-00499-t007:** The content of FAs in the eggs (% weight of FAs in the sample) after 90 days of the experiment.

Fatty Acids	Control	Formulation of SC-Extract	Microalga—*Spirulina Platensis*	Residue	Formulation without SC-Extract	SEM	*p*-Value
12:0	0.0178	0.0161	0.0213	0.0213	0.0196	0.00089	0.612
14:0	0.328	0.304	0.329	0.341	0.312	0.0100	0.796
14:1 (n-5)	0.0430 ^AB^	0.0449 ^AB^	0.0479 ^AB^	0.0547 ^A^	0.0365 ^B^	0.00164	0.009
15:0	0.0579	0.0548	0.0564	0.0566	0.0582	0.00157	0.745
16:0	22.0	21.8	22.3	22.7	21.9	0.506	0.616
16:1 (n-7)	2.28	2.19	2.21	2.36	2.04	0.057	0.329
17:0	0.164	0.160	0.168	0.162	0.172	0.0040	0.718
18:0	6.77	6.97	7.04	7.03	6.97	0.143	0.445
18:1 (n-9)	38.5	38.7	36.7	36.6	38.6	0.782	0.840
18:2 (n-6)	13.5	13.6	15.2	15.0	13.9	0.320	0.058
18:3 (n-6)	0.094	0.096	0.095	0.111	0.099	0.0027	0.304
18:3 (n-3)	0.515 ^A^	0.495 ^A^	0.311 ^B^	0.306 ^B^	0.532 ^A^	0.0146	0.001
19:0	0.0411 ^a^	0.0472 ^ab^	0.0512 ^ab^	0.0544 ^b^	0.0456 ^ab^	0.00147	0.041
20:1 (n-9)	0.192	0.214	0.200	0.185	0.204	0.0044	0.269
20:2 (n-6)	0.092 ^A^	0.097 ^A^	0.126 ^Ba^	0.118 ^ABab^	0.100 ^b^	0.0032	0.001
20:3 (n-3)	0.0996	0.104	0.106	0.107	0.101	0.0024	0.850
20:4 (n-6)	1.44	1.43	1.52	1.53	1.47	0.030	0.421
20:5 (n-3)	0.328	0.412 ^a^	0.366	0.349	0.271^b^	0.0150	0.034
22:5 (n-3)	0.301	0.339 ^a^	0.272^b^	0.285	0.288	0.0073	0.042
22:6 (n-3)	0.646 ^A^	0.629 ^A^	0.415 ^B^	0.440 ^B^	0.669^A^	0.0168	0.001
SFA	29.3	29.4	30.1	30.4	29.5	0.658	0.459
MUFA	41.1	41.2	39.2	39.2	40.9	0.835	0.892
PUFA	17.1	17.2	18.4	18.2	17.4	0.380	0.313
PUFA n-3	1.89 ^A^	1.98 ^A^	1.47 ^Ba^	1.49 ^Ba^	1.86 ^b^	0.044	0.001
PUFA n-6	15.2 ^a^	15.2 ^a^	16.9 ^b^	16.7 ^ab^	15.6 ^ab^	0.352	0.046
n-6/n-3	8.08 ^A^	7.70 ^A^	11.59 ^B^	11.3 ^B^	8.44 ^A^	0.204	0.001
PUFA/SFA	0.585	0.586	0.617	0.603	0.595	0.0054	0.335

SEM—standard error of measurements; SFA—saturated fatty acids; MUFA—monounsaturated fatty acids; ^a, b^—statistically significant differences between the groups for individual parameters, at *p* < 0.05.; ^A, B^—statistically significant differences between the groups for individual parameters, at *p* < 0.01.

**Table 8 animals-10-00499-t008:** The content of FAs in the eggs (% weight of FAs in the sample) after 120 days of the experiment.

Fatty Acids	Control	Formulation of SC-Extract	Microalga—*Spirulina Platensis*	Residue	Formulation without SC-Extract	SEM	*p*-Value
12:0	0.00538 ^A^	0.00895 ^ABab^	0.00536 ^a^	0.00534 ^a^	0.0106 ^Bb^	0.00060	0.005
14:0	0.295	0.275	0.269	0.303	0.286	0.0081	0.562
14:1 (n-5)	0.047	0.046	0.046	0.577	0.560	0.0018	0.216
15:0	0.0630 ^a^	0.0615 ^ab^	0.0500 ^b^	0.0563 ^ab^	0.0515 ^ab^	0.00170	0.049
16:0	22.1	21.7	22.0	22.2	21.9	0.492	0.886
16:1 (n-7)	2.18	2.10	2.12	2.37	2.29	0.060	0.589
17:0	0.176	0.181	0.159	0.162	0.156	0.0041	0.172
18:0	6.91	7.24	7.41	6.75	6.87	0.146	0.091
18:1 (n-9)	38.1	37.8	38.3	38.7	38.4	0.776	0.695
18:2 (n-6)	14.4	14.9	13.5	12.9	13.7	0.323	0.229
18:3 (n-6)	0.0917	0.101	0.0902	0.0943	0.0973	0.00225	0.671
18:3 (n-3)	0.420	0.419	0.370	0.428	0.404	0.0102	0.238
19:0	0.0720	0.0795	0.0749	0.0658	0.0732	0.00187	0.228
20:1 (n-9)	0.211	0.228	0.225	0.214	0.224	0.0045	0.702
20:2 (n-6)	0.117	0.130	0.124	0.113	0.118	0.0031	0.504
20:3 (n-3)	0.106	0.119	0.113	0.106	0.102	0.0024	0.171
20:4 (n-6)	1.49	1.54	1.53	1.50	1.47	0.030	0.906
20:5 (n-3)	0.537	0.581	0.606	0.517	0.549	0.0137	0.247
22:5 (n-3)	0.274 ^a^	0.304 ^ab^	0.333 ^b^	0.269 ^a^	0.308 ^ab^	0.0075	0.039
22:6 (n-3)	0.547	0.568	0.468	0.509	0.510	0.0114	0.055
SFA	29.6	29.5	30.0	29.6	29.3	0.642	0.884
MUFA	40.5	40.1	40.7	41.1	41.0	0.830	0.629
PUFA	18.0	18.7	17.2	16.5	17.3	0.387	0.235
PUFA n-3	1.88	1.99	1.89	1.83	1.87	0.038	0.628
PUFA n-6	16.1	16.7	15.3	14.6	15.4	0.353	0.250
n-6/n-3	8.61	8.39	8.12	8.06	8.23	0.101	0.195
PUFA/SFA	0.612 ^AB^	0.637 ^A^	0.577 ^AB^	0.561 ^B^	0.591 ^AB^	0.0067	0.008

SEM—standard error of measurements; SFA—saturated fatty acids; MUFA—monounsaturated fatty acids; ^a, b^—statistically significant differences between the groups for individual parameters, at *p* < 0.05.; ^A, B^—statistically significant differences between the groups for individual parameters, at *p* < 0.01.

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
