# Peer review of "Biofortification of Hens Eggs with Polyunsaturated Fatty Acids by New Dietary Formulation: Supercritical Microalgal Extract"

_animals, 2020, doi:10.3390/ani10030499_

Round 1

Reviewer 1 Report

It is important to justify why 6 replicates per group (treatment) were used only.

There are several english redaction details that should be improved (some are already highlighted in file)

Table 6 is the most important then it must be improved reducing the information it contains. Actually there is not information regarding statistical analysis from results in that table   

Author Response

Thank you so much for your valuable review. Below are the answers to your comments.

It is important to justify why 6 replicates per group (treatment) were used only.

This is explained in the text now. „This layout of the experiment is due to the fact that the battery system is located in the middle of room. The cages are located on the right and left side of the battery system, Each replication consisted of cages from the first second and third floor on the right and left side of the battery system. This layout best reflects the conditions in the henhouse”.

There are several english redaction details that should be improved (some are already highlighted in file).

I have corrected the language errors that you have marked in the text.

Table 6 is the most important then it must be improved reducing the information it contains. Actually there is not information regarding statistical analysis from results in that table.

Table 6 has been replaced by 4 smaller ones. SD has been replaced by SEM. The tables also provide a p-value and statistically significant differences.

Reviewer 2 Report

General comments:
1. A major concern in this study is the addition of these feed additives in the water. Firstly, the authors must measure the water consumption to determine the actual amount of feed additives reach each hen. How the authors be sure that the proper amount of feed additives were given to each hen? What about the solubility of these additives in the water? What is the drinking method used nipple or potteries?
2. Justification of the doses used of the feed additives is highly recommended to be clarified with the appropriate references.
3. The presentation of the tables really needs to be reorganized. For instance, the tables are very crowded thus the vertical presentation is preferable than horizontal.
4. There is a problem in using the abbreviations throughout the manuscript. The abbreviation must be introduced upon the first mentioning of the full term followed by its abbreviation in parentheses: From then on, the abbreviation must be used exclusively and throughout.
5. The manuscript contains many grammatical, typographic, and styling errors. Revision of the manuscript by a native English speaker is highly recommended. Also, the SEM column is enough thus deleting SD columns are better. Importantly, the letters of significance (a, b, c, …..etc) are highly recommended to be added to all tables.
6. Why the author used the formulation without extract group containing the emulsifier only without the other two ingredients (i.e. Potassium sorbate (Preservative) and Citric acid (Antioxidant) ) in the products
Specific comments:
1. Abstract: the results need to be presented in a more organized manner and the conclusion is missed.
2. Introduction:
- The first paragraph (line 41-46) needs to be supported with a reference.
- Line 50: what "they" refer to?
- Line 52-56: very long sentence. Please, rephrase.
- Line 81-102: need to be more concise or deleted.
- The hypothesis of the study should be clarified before the aim.
3. Material and methods:
- Line 140: delete "preparation of".
- Line 147: replace with "animal diets and experimental design".
- Lines 169-170: transfer to lines 148-150. Then, the appropriate reference should be added.
4. Results and discussion:
- Begin the results with the fatty acid profile of tested additives followed by production parameters.
- The feed intake should be clearly discussed.
- Lines 235-243: begin the results then discuss it
- Table 2: the nutrient composition of the basal diet should be added. Data in table 3 should be added as a footnote in table 2 and table 3 should be removed.
- In table 4, three parameters should be added including egg weight, egg mass, and FCR.

Author Response

Thank you so much for your valuable review. Below are the answers to your comments.

General comments:

A major concern in this study is the addition of these feed additives in the water. Firstly, the authors must measure the water consumption to determine the actual amount of feed additives reach each hen. How the authors be sure that the proper amount of feed additives were given to each hen? What about the solubility of these additives in the water? The additives were well soluble in water.
What is the drinking method used nipple or potteries? The reference point was the daily dose of algae powder recommended by the literaturę. Ab extract was made from this dose and this amount was given to hens.

Liter droplets were used in the experiment. The doses were developed based on recommendations on S. platensis supplementation. We focused on administering the right volume of additive to the volume of water, so the hens should receive the right dose of preparations.

Justification of the doses used of the feed additives is highly recommended to be clarified with the appropriate references.

The doses were developed based on recommendations on S. platensis supplementation.

The presentation of the tables really needs to be reorganized. For instance, the tables are very crowded thus the vertical presentation is preferable than horizontal.

We stayed with the horizontal presentation, however SD was removed from the tables and only SEM and p-value were left. Thanks to this, the tables are smaller, less crowded and more readable.

There is a problem in using the abbreviations throughout the manuscript. The abbreviation must be introduced upon the first mentioning of the full term followed by its abbreviation in parentheses: From then on, the abbreviation must be used exclusively and throughout.

Your suggestion has been applied.

The manuscript contains many grammatical, typographic, and styling errors. Revision of the manuscript by a native English speaker is highly recommended. Also, the SEM column is enough thus deleting SD columns are better. Importantly, the letters of significance (a, b, c, …..etc) are highly recommended to be added to all tables.

SD has been replaced by SEM. The letters of significance has been added. English has been improved.

Why the author used the formulation without extract group containing the emulsifier only without the other two ingredients (i.e. Potassium sorbate (Preservative) and Citric acid (Antioxidant) ) in the products

The preservative and antioxidant were added to the extract only to maintain its stability throughout the experiment. The used additives were potassium sorbate and citric acid, they could not affect the fatty acid profile of the egg.

Specific comments:

Abstract: the results need to be presented in a more organized manner and the conclusion is missed.

The results were organized. Conclusions have been added.

Introduction:

- The first paragraph (line 41-46) needs to be supported with a reference.

This paragraph has been supported with a reference.

- Line 50: what "they" refer to?

They changed to PUFAs. However, as recommended by another reviewer, this paragraph has been removed.

- Line 52-56: very long sentence. Please, rephrase.

Your suggestion has been applied. „The market offers many products that are industrially fortified with n-3 and n-6 fatty acids (which differ in the location of the first double bond between the carbon atoms). These products are for example oils, mayonnaises, margarines, salad dressings, bakery products, infant formula, milk, meat and poultry products, farmed fish and eggs”.

- Line 81-102: need to be more concise or deleted.

This part of the paper has been deleted.

- The hypothesis of the study should be clarified before the aim.

The hypothesis have been clarified before the aim. „Algae biomass is a rich source of fatty acids. However, it contains fiber that can be a limiting factor in poultry nutrition. Therefore, it is better to use of algae extracts. Due to the fact that the extracts have a liquid form, it seems that it is better to use them as supplements for a drinking water rather than for feed. Emulsifiers are added to increase the solubility of the extracts in water. However, emulsifiers can affect the results of the experiment. The supercritical extraction process also produces a large amount of residue, which can also contain biologically active compounds in its biomass”.

Material and methods:

- Line 140: delete "preparation of".

Your suggestion has been applied.

- Line 147: replace with "animal diets and experimental design".

Your suggestion has been applied.

- Lines 169-170: transfer to lines 148-150. Then, the appropriate reference should be added.

In my opinion this part of the manuscript should remain in place.

Results and discussion:

- Begin the results with the fatty acid profile of tested additives followed by production parameters.

Your suggestion has been applied.

- The feed intake should be clearly discussed.

Feed intake has been replaced by FCR and clearly discussed.

- Lines 235-243: begin the results then discuss it

In my opinion, this part of the manuscript is a good introduction to presenting and discussing the results.

- Table 2: the nutrient composition of the basal diet should be added. Data in table 3 should be added as a footnote in table 2 and table 3 should be removed.

Data from table 3 has been added as a footnote in table 2 and table 3 has been removed.

- In table 4, three parameters should be added including egg weight, egg mass, and FCR

FCR has been added. Egg weight and egg mass will be presented in another article in which we will focus on the production and health parameters of the birds. For now, we would prefer not to present them, especially since these results were not the aim of the present study.

Reviewer 3 Report

This manuscript deals with the effect of adding different algal products (biomass, algal extracts, residue or emulsifier) on laying hens production and yolk fatty acid profile. The novelty of the manuscript is the way authors use to provide treatmens: drinking water. Although the effects are not significant for all the measured variables authors report and discuss results as if they were significant. Try not to treat numerical differences and trends to differences as being equivalent to statistical differences. Therefore, my suggestion is to reject the manuscript as it is.

General comments

The introduction should be greatly improved. Authors feed extracts, biomass, the residue and the emusifier. It is not clear why some of these treatments are given. Why should the emulsifier or the residue be fed? What is the advantage. Treatmens are given in the drinking water. Why? This is the novelty of the manuscript. What is the advantage? Clearly describe the hypothesis. 

The objective needs to be reworded. Authors compared and discussed the effect of treatment day. This should be included.

The results are described and discussed as if authors had found significant differences. Always be certain that the statistical model is consistent with your objectives and that your statements relative to treatment differences are statistically fully supported. Statements based upon visual appraisal of data (e.g., ‘there was a numerical difference between treatment A and
B’) cannot be accepted.

In order to make comparisons with other manuscripts and to imrove the understanding of the results obtained in the current trial authors should provide: 1) the ether extract and the FA profile of the basal diet, and 2) an estimation of water intake because treatmens are provided here. With this information readers could have an idea of the relative input of fatty acid in water compared to that in the basal diet.

Conclusions need to be reworded. The ‘Conclusions’ should be directly connected to the objectives and hypothesis statements (i.e., indicate how the objectives were met in full or in part). They should both summarize the key findings of the study and indicate their implications, and nothing else.

Specific comments

Line 30-35. Report significant or non significant effects, not numerical values. A conclusion is missing.

Line 47. Add reference

Line 48-51. Delete

Line 116. Be more specific. What is "tested"?

Line 119-120. Delete

Line 152-156. Add information on layers age at the beginning of the trial. Duration of the trial. How was feed and water provided? How were treatments added to water? Merge Table 2 and 3. Provide information on chemical composition and fatty acid profile. "Ingredients" instead of "composition". Are ingredients as fed or on dry matter basis? Layers were not supplemented with lysine and methionine?

Line 166-167. "at the end of each series" (?)

Line 167-168. This is discussion

Line 185 and elsewhere. According to Table 5, did authors take 2 samples? If so, be more specific here.

Table 4. "laying index" instead of "egg production". Did authors collect eggs daily? it is not reported in the material and methods section. Report feed intake in g/d. Is it dry matter intake? How did authors measure feed intake? not reported in the material and methods section. It is quite surprising that authors did not find significant differences between control and sc-extract (95 vs 87%) with a rather low SEM. Report the feed convertion ratio.

Line 220-221, 222-223 and elsewhere. But are these differences significant?

Line 231-233. Repeated idea.

Line 234. Authors should include this in the introduction and objectives. Otherwise this should be included in the material and methods section as a supplementary table.

Line 236-237. Compared to? on what grounds?

Line 252-254, and compared to the basal diet on g/d?

Line 272-273. Not only of control group. The other groups are fed with the basal diet as well.

Line 274-275. Adaptation to a change in the feed fatty acid profile occurs in less than two weeks as you mention in the discussion. Please reword or delete.

Table 6. Why are resutls after 30 days reported on the basis of 5 replicates and after 60 and 120 on 6 replicates? Table is not fully seen.

Line 322-323. On what grounds?

Line 336-349, are differences significant? Did authors analyze the effect of the day? How? not reported in the material and methods section.

Author Response

Thank you so much for your valuable review. Below are the answers to your comments.

This manuscript deals with the effect of adding different algal products (biomass, algal extracts, residue or emulsifier) on laying hens production and yolk fatty acid profile. The novelty of the manuscript is the way authors use to provide treatmens: drinking water. Although the effects are not significant for all the measured variables authors report and discuss results as if they were significant. Try not to treat numerical differences and trends to differences as being equivalent to statistical differences. Therefore, my suggestion is to reject the manuscript as it is.

General comments

The introduction should be greatly improved. Authors feed extracts, biomass, the residue and the emusifier. It is not clear why some of these treatments are given. Why should the emulsifier or the residue be fed? What is the advantage. Treatmens are given in the drinking water. Why? This is the novelty of the manuscript. What is the advantage? Clearly describe the hypothesis. 

The introduction has been improved. The hypothesis has been described. „Algae biomass is a rich source of fatty acids. However, it contains fiber that can be a limiting factor in poultry nutrition. Therefore, it is better to use of algae extracts. Due to the fact that the extracts have a liquid form, it seems that it is better to use them as supplements for a drinking water rather than for feed. Emulsifiers are added to increase the solubility of the extracts in water. However, emulsifiers can affect the results of the experiment. The supercritical extraction process also produces a large amount of residue, which can also contain biologically active compounds in its biomass”.

The objective needs to be reworded. Authors compared and discussed the effect of treatment day. This should be included.

The objective has been reworded.

The results are described and discussed as if authors had found significant differences. Always be certain that the statistical model is consistent with your objectives and that your statements relative to treatment differences are statistically fully supported. Statements based upon visual appraisal of data (e.g., ‘there was a numerical difference between treatment A and B’) cannot be accepted.

After improvements, the tables highlighted statistically significant results and discussed again.

In order to make comparisons with other manuscripts and to imrove the understanding of the results obtained in the current trial authors should provide: 1) the ether extract and the FA profile of the basal diet, and 2) an estimation of water intake because treatments are provided here. With this information readers could have an idea of the relative input of fatty acid in water compared to that in the basal diet.

Fatty acid profile of the basal diet has not been studied. All birds received exactly the same feed, so the differences in fatty acid profile in eggs could only be due to the additives given in the drinking water.

Conclusions need to be reworded. The ‘Conclusions’ should be directly connected to the objectives and hypothesis statements (i.e., indicate how the objectives were met in full or in part). They should both summarize the key findings of the study and indicate their implications, and nothing else.

Your suggestion has been applied. Conclusions has been reworded.

Specific comments

Line 30-35. Report significant or non significant effects, not numerical values. A conclusion is missing.

Significant effects, non-significant effects and conclusions have been added.

Line 47. Add reference

Reference has been added.

Line 48-51. Delete

Your suggestion has been applied.

Line 116. Be more specific. What is "tested"?

It was noted that it was about formulation of super critical extract, microalgae Spirulina platensis, post-extraction residue and formulation without extract (containing only an emulsifier).

Line 119-120. Delete

Your suggestion has been applied.

Line 152-156. Add information on layers age at the beginning of the trial. Duration of the trial. How was feed and water provided? How were treatments added to water? Merge Table 2 and 3. Provide information on chemical composition and fatty acid profile. "Ingredients" instead of "composition". Are ingredients as fed or on dry matter basis? Layers were not supplemented with lysine and methionine?

The information on layers age has been added at the beginning of the trial. Duration of the trial has been added. Information on providing water and feed is included in the manuscript. Tables 2 and 3 have been merged. Composition changed to ingredients. Ingredients were fed. Information on fatty acid profile in additives is contained in the manuscript. Lysine and methionine were not supplemented. This is because we wanted the feed mixture to be as simple as possible.

Line 166-167. "at the end of each series" (?)

It was noted that eggs for fatty acids analysis were collected after 30, 60, 90 and 120 days of the experiment.

Line 167-168. This is discussion

This part of the manuscript has been removed.

Line 185 and elsewhere. According to Table 5, did authors take 2 samples? If so, be more specific here.

Thank you for that remark. It has been specified.

Table 4. "laying index" instead of "egg production". Did authors collect eggs daily? it is not reported in the material and methods section. Report feed intake in g/d. Is it dry matter intake? How did authors measure feed intake? not reported in the material and methods section. It is quite surprising that authors did not find significant differences between control and sc-extract (95 vs 87%) with a rather low SEM. Report the feed convertion ratio.

Egg production changed to laying index. Feed intake was replaced by feed convertion ratio. Eggs were collected daily, it was written in materials and methods. I’m sure that statistical analysis showed no significant differences in the laying index.

Line 220-221, 222-223 and elsewhere. But are these differences significant?

The results and discussion has been rewritten.

Line 231-233. Repeated idea.

This part of the manuscript has been deleted.

Line 234. Authors should include this in the introduction and objectives. Otherwise this should be included in the material and methods section as a supplementary table.

It has been included in the objectives.

Line 236-237. Compared to? on what grounds?

Should be: It was noticed that supercritical  fluid extraction extraction favored isolation of saturated and unsaturated fatty acids when compared to traditional extraction techniques with the use of organic solvents.

Line 252-254, and compared to the basal diet on g/d?

It was secified that it was microalgae, residue and formulation without SC-extract.

Line 272-273. Not only of control group. The other groups are fed with the basal diet as well.

This part of the manuscript has been removed.

Line 274-275. Adaptation to a change in the feed fatty acid profile occurs in less than two weeks as you mention in the discussion. Please reword or delete.

The results and discussion have basically been rewritten from the beginning.

Table 6. Why are results after 30 days reported on the basis of 5 replicates and after 60 and 120 on 6 replicates? Table is not fully seen.

Thank you for that remark. It was a mistake. All results were reported on the basis of 5 replicates.

Line 322-323. On what grounds?

Results and discussion have been reworded.

Line 336-349, are differences significant? Did authors analyze the effect of the day? How? not reported in the material and methods section.

Of course you have right. We performed a one-way analysis of variance, so the effect of the day was not included in this particular analysis. In this part, we focused on the occurrence of trends, it seems to us sufficient.

Round 2

Reviewer 2 Report

The manuscript has greatly improved, but some concerns need to be fixed before publication as follows:

  • Line 23: Which Fatty acids? mention these FAs briefly.
  • Please minimize the Abstract section to 200 words.
  • Please carefully revise the use of abbreviations throughout the manuscript. The abbreviation must be introduced upon the first mentioning of the full term followed by its abbreviation in parentheses: From then on, the abbreviation must be used exclusively and throughout.
  • Line 128: delete "preparation of".
  • Table 2: the calculated or analysed chemical composition of the basal diet should be added.
  • In all tables, uniform the use of the letters of significance (a, b, c, …..etc) to be capital or small letters. Also, in Table 7 (19:0) the significance letters are missed.
  • Table 8 is not in the correct place.

Author Response

Thank you so much for your valuable revision. Below are the answers for your comments.

The manuscript has greatly improved, but some concerns need to be fixed before publication as follows:

  • Line 23: Which Fatty acids? mention these FAs briefly.

These fatty acids have been mentioned. „Analysis of results obtained during tests showed that used additives may increase the content of such fatty acids in eggs as: docosapentaenoic acid (DPA, C22:5 n-3), eicosadienoic acid (C22:2 n-6), and the total omega-6 polyunsaturated fatty acids content (PUFA n-6). The used additives may also reduce the content of saturated fatty acids (SFA) such as: pentadecanoic acid (C15:0).”

  • Please minimize the Abstract section to 200 words.

Your suggestion has been applied. The abstract has been shortened and now has 189 words.

  • Please carefully revise the use of abbreviations throughout the manuscript. The abbreviation must be introduced upon the first mentioning of the full term followed by its abbreviation in parentheses: From then on, the abbreviation must be used exclusively and throughout.

The use of abbreviations throughout the manuscript has been revised.

  • Line 128: delete "preparation of".

Your suggestion has been applied.

  • Table 2: the calculated or analysed chemical composition of the basal diet should be added.

Chemical composition of basal diet has been added.

  • In all tables, uniform the use of the letters of significance (a, b, c, …..etc) to be capital or small letters. Also, in Table 7 (19:0) the significance letters are missed.

Thank you for this comment, but we would like to keep in Tables capital and small letter. Small letters were used to show statistically significant differences between the groups for individual parameters for P<0.05, whereas capitals to show statistically significant differences between the groups for individual parameters for P<0.01. This information is included for each Table. This is a popular manner used in papers to distinguish the significance of differences. Significance for acid (19:0) was determined.

  • Table 8 is not in the correct place.

Thank you for this comment. We see that in the pdf file, this Table has shifted, but in the Word file which was sent to the Journal it was in the correct place. Probably this is due to the system fault. Now, after generation of pdf file, Table 8 is in the correct place.

Reviewer 3 Report

The manuscript has been improved, but still requieres further improvements.

General comments

Authors focus on showing what treatment had the greatest or the lowest value, but authors should focus on showing only significant differences among treatments.

But not only, special emphasis should be given to differences between control and extract. What is the advantage of feeding any treatment if it is not different from the actual feeding practice (Control)? 

Conclusion. Reword conclusion according to at least the response compared to the control.

Specific comments

Line 83-85. Processing conditions are not relevant at this point. Please delete.

Line 97. Oils commonly fed are also in liquid form. The one authors used, soybean oil, is in liquid form. Any other advantage?

Line 140. Provide information on the initial laying index, before the beginning of the trial.

Line 143. Unless authors provide the chemical composition on Table 2: energy, crude protein, aminoacids, readers cannot check.

Line 165. Add reference or provide more details.

Table 4 and any other result Table. Round to 0 decimals the laying index value. Provide SEM with one more decimal than that used for the variable. P value with 3 decimals. Laying index is extremely affected the low number of hens used in the trial. One hen not laying out of three in each replicate has a tremendous impact on the laying index. FCR: only superscripts given for SC-extract and without SC-extract, is control different from SC-extract? Provide all superscrips when authors find a significant treatment effect. 

Line 268-269. compared to what treatment? to control? Either delete or reword.

Table 6 and any other result table. Report superscripts only for P<0.05. It is very confusing as it is now.

Line 354-363. Is this one of the objectives? if so, clearly state as an objective in the introduction and specify how authors did this comparison in the statistical analysis section.

Line 398-399, line 404-405. Compared to control?

Line 399-402. Please delete

Line 407-409. This is discussion, not a conclusion

Author Response

Thank you so much for your valuable revision. Below are the answers for your comments.

The manuscript has been improved, but still requieres further improvements.

General comments

Authors focus on showing what treatment had the greatest or the lowest value, but authors should focus on showing only significant differences among treatments.

But not only, special emphasis should be given to differences between control and extract. What is the advantage of feeding any treatment if it is not different from the actual feeding practice (Control)? 

Conclusion. Reword conclusion according to at least the response compared to the control.

We think that comparing different treatments is important because it shows that the results of the extract and emulsifier groups overlap. In presenting the results, we focused on comparing different treatment methods between ourselves and control. The conclusions also contain information on which treatment has the greatest impact on the fatty acid content compared to the control. It was also written in which acids the differences are significant.

Specific comments

Line 83-85. Processing conditions are not relevant at this point. Please delete.

Processing conditions have been deleted.

Line 97. Oils commonly fed are also in liquid form. The one authors used, soybean oil, is in liquid form. Any other advantage?

In the manuscript, there is information concerning additional advantages of the liquid form of the additive: „Due to the fact that the extracts have a liquid form, it seems that it is better to use them as supplements for a drinking water which was allowed by the developed formulation. In addition, the extract used an additive to feed could be exposed to high temperatures during its production. This could have a negative impact on the biologically active compounds contained in the extract.”

Line 140. Provide information on the initial laying index, before the beginning of the trial.

As you suggested, the following information is at work„The laying index of hens in the time of insertion was 91%”.

Line 143. Unless authors provide the chemical composition on Table 2: energy, crude protein, aminoacids, readers cannot check.

Chemical composition of basal diet has been added to this Table.

Line 165. Add reference or provide more details.

“When choosing the doses of extracts that were investigated during the study, the economic criterion was mainly taken into account, i.e. whether the cost of producing the additive (costs of obtaining the raw material, its extraction and formulation production) would be rational for farmers”. This was written in the manuscript.

Table 4 and any other result Table. Round to 0 decimals the laying index value. Provide SEM with one more decimal than that used for the variable. P value with 3 decimals. Laying index is extremely affected the low number of hens used in the trial. One hen not laying out of three in each replicate has a tremendous impact on the laying index. FCR: only superscripts given for SC-extract and without SC-extract, is control different from SC-extract? Provide all superscrips when authors find a significant treatment effect. 

Laying index has been rounded to 0. SEM has one more decimal than variables. The P-value has 3 decimals. Superscripts at FCR have been assigned to all groups. Control is not significantly different from SC-extract.

Line 268-269. compared to what treatment? to control? Either delete or reword.

Thank you for that remark. The sentence, of course, was incorrect and was reworded. „Based on the results, it can be concluded that the used additives had no significant effect on bird production parameters.”

Table 6 and any other result table. Report superscripts only for P<0.05. It is very confusing as it is now.

Thank you for this comment, but we would like to keep in Tables capital and small letter. Small letters were used to show statistically significant differences between the groups for individual parameters for P<0.05, whereas capitals to show statistically significant differences between the groups for individual parameters for P<0.01. This information is included for each Table. This is a popular manner used in papers to distinguish the significance of differences.

Line 354-363. Is this one of the objectives? if so, clearly state as an objective in the introduction and specify how authors did this comparison in the statistical analysis section.

This was not one of the main objectives of the research. Therefore, this part of the manuscript has been removed.

Line 398-399, line 404-405. Compared to control?

As it was shown above, the used additives did not affect the production parameters of birds. Accordingly, the sentence has been reworded and now reads: “The obtained results suggest that supplementation of the additives used in the experiment has no effect on the production parameters of laying hens”. Changes in the fatty acids profile obviously refer to comparison with the control group, this was written in the manuscript.

Line 399-402. Please delete

You suggestion has been applied.

Line 407-409. This is discussion, not a conclusion

This part of the manuscript has been deleted.
